# Towards an Understanding of Successes of the Psychiatric Nurses in Caring for Children with Mental Health Problems: An Appreciative Inquiry

**DOI:** 10.3390/ijerph20031725

**Published:** 2023-01-18

**Authors:** Rorisang Mary Machailo, Daleen Koen, Molekodi Matsipane

**Affiliations:** School of Nursing, Faculty of Health Sciences, North West University, Mahikeng 2735, South Africa

**Keywords:** appreciative inquiry, mental health, psychiatric nurses, success

## Abstract

Introduction: Psychiatric nurses have a specialized body of knowledge and skills in providing care to persons with mental health challenges. The literature provides scanty evidence on child psychiatric nursing practices. This paper explored the successes of psychiatric nurses in caring for children with mental health problems using appreciative inquiry (AI). Design: A qualitative exploratory and descriptive design was used to allow for new ideas that can fundamentally reshape the practice of child psychiatric nursing. Purposive sampling was used to select psychiatric nurses caring for children with mental health problems. Focus groups were used to generate data. Findings: The results indicate both positive and negative prospects for psychiatric nursing practice. The positive possibilities included commitment, passion and dedication of staff to the children. The negative aspects that need urgent attention include lack of specific, integrated child mental health within the mental health care services, shortage of resources and not-fit-for purpose infrastructure. Conclusion: Appreciative inquiry verified the commitment of psychiatric nurses in caring for children with mental health problems and the potential for dedicated child psychiatric institutions in realizing the needs of such children. The needs of children with mental health problems must be addressed through positive care in the health system.

## 1. Introduction

Psychiatric nursing has become more complex because it has evolved into an interpersonal process reflecting the reciprocity in human relationships. It is a caring, reflective, and therapeutic practice that promotes recovery and emotional health [1]. Psychiatric nurses have a specialized body of knowledge and skills that are essential in providing care to persons with mental health challenges [2]. It is the authors’ view that psychiatric nursing creates change and fosters positive knowledge about the self. Caring is one of the core values that distinguishes psychiatric nursing from other nursing disciplines. Previous studies have shown that caring happens within a relationship between a nurse and a patient [3].

Psychiatric nurses caring for children with mental illness in North West Province (NWP) in South Africa work across a wide variety of settings. Within mental health care institutions, they are found in psychiatric emergency departments, acute psychiatric inpatient units, rehabilitation units, long-term care, and outpatient clinics. They can also be found in general health care settings, such as emergency rooms, which have become the front line of treatment for acute psychiatric presentations, despite their lack of adequate resources to deal with these problems. Psychiatric nurses also have presence in pediatric units and pediatric outpatient clinics. In the community, they work largely in primary health care services, such as community health centers and clinics [3].

The researchers in this study view success as the opportunity to expand and influence the nursing practice environment. Reference [4] defined success as the accomplishment of a set goal because one succeeds whenever one is able to translate a set goal into reality. It is important, therefore, for psychiatric nurses to identify critical success factors in caring for children with mental health problems. According to [5], factors which assist psychiatric nurses to feel positive about themselves include therapeutic success, good teamwork, targeted education and feedback. Therefore, if psychiatric nurses can relate therapeutically to the children they care for, this would demonstrate efficiency and success in the care provided.

The mental health care system in NWP is nurse-based because the geospatial characteristics are largely rural. The current child psychiatric service in this province is scanty and not well defined. There are currently no psychiatric specialists in the rural communities, except in the specialized psychiatric hospitals and health care institutions, where dedicated child psychiatric services are offered. Psychiatric nurses are, therefore, expected to be competent and have the required expertise to offer quality mental health care to everyone in need of it in their unique environment. Psychiatric nurses in this province constitute the largest group of health care professionals providing for mental health care users in accredited psychiatric health care services [6]. Even though the psychiatric nurse’s role is often underestimated and under-reported, these nurses are highly trusted [2]. Generally, the psychiatric nursing profession (particularly those caring for children with mental health challenges) does not excel at promoting its own interests [2]. As such, there is currently no evidence in the literature that focuses on child mental health nursing in NWP. However, psychiatric nurses still need to take care of children with mental health problems due to the high demand for such services. Furthermore, there is a scarcity of information on child mental health services in NWP.

Children are generally vulnerable, and the manifestation of mental illness complicates the situation more. Reference [7] highlighted that the emotional wellbeing of children is as important as their physical health, and both must be prioritized. Notwithstanding, it is also important that the health care system throughout the province prioritize children’s mental health the same way they do physical health. Reference [7] recommended that the children must be enabled to achieve their full developmental potential, and this serve as a human right and an essential requisite for sustainable development. The sustainable development goals (SDGs) also explicitly include ensuring healthy lives and promoting well-being for all persons including children. According to [8], more studies have confirmed that among the main factors for healthy and successful child development, child mental health is of paramount importance.

Psychiatric nurses in clinical practice commonly encounter children with mental health challenges who often are unaware that they have behavioral or emotional problems. Caring for this population is complex and can be overwhelming for the inexperienced mental health care provider [9,10]. The literature highlights that to provide adequate and quality care, psychiatric nurses require specialized child psychiatric nursing knowledge and skills to care for children with mental health challenges [6]. Nursing schools have a role to play in emphasizing the importance of psychiatric nurses’ contribution to health care and preparing nurses for new roles in clinical practice [2]. From an array of possibilities, this paper proposes appreciative inquiry (AI) as the perspective that best articulates an understanding of successes of psychiatric nurses in caring for children with mental health problems.

Despite little research in the psychiatric field in NWP in South Africa, a few qualitative inquiries undertaken outside this province thus far have highlighted the need to reconsider whether psychiatric nurses have sufficient knowledge and skill to deal with the complexities of child mental health nursing [6]. To help fill this gap, the study dealt with AI, focusing on critical success factors expressed as attributes in understanding the psychiatric nurses in caring for children with mental health problems [11]. This study contributes to the psychiatric nursing practice by identifying opportunities for children with mental health problems in NWP.

## 2. Materials and Methods

### 2.1. Study Design

In this study, the authors draw from participatory action research (PAR) as a mechanism for interventions and development in psychiatric nursing community. We highlight the importance of rethinking the basic assumptions embedded in PAR method as a mode of action research for psychiatric nurses caring for children with mental health challenges. Appreciative Inquiry (AI), as a growing practice in organizational development, is presented as an opportunity for change which encourages innovation and collaboration through participatory methods [12]. This study is about extending and elevating strengths within the psychiatric nursing practice.

AI was used in this study as a philosophy, process, and method for promoting transformational change, shifting from a traditional problem-based orientation to a strength-based approach. The intention is to change and focus on affirmation, appreciation, and positive dialogue [13]. However, it is a complex intervention, because it challenges the dominant hierarchical power relation and empowers practitioners to become change agents who explore innovative practice [14]. Reference [15] indicated that AI supports learning and reflection in a positive way. This AI was focused on the success of psychiatric nurses, the strength and their best practices in caring for children with mental health problems in NWP.

### 2.2. Appreciative Inquiry (AI) Assumptions

AI methodology is adaptable and flexible, promotes resilience and contributes to the development of professional identity in a safe environment in its implementation [10]. AI has been reported as useful in supporting change in nursing practice [16]. Tapping into the motivations for change and what gives life a meaning using a positive approach can unlock the collective intelligence and construct positive team capacity [12]. This methodological approach invites the psychiatric nurses to see themselves and the practice environment through an appreciative eye, an eye that values possibility.

The literature describes AI as a focus on best experience and affirms the best qualities within any living system. Appreciative inquiry has been successfully used as a research strategy to facilitate practice change. Reference [17] indicated that AI has been reported as a catalyst for practice change, emphasizing collaboration in research and the development of practices. At the same time, Ref. [18] indicated that there is little empirical evidence to support the effectiveness of AI in improving healthcare but that it may have a positive impact on clinical care. AI was used in this study to better understand the successes of psychiatric nurses in caring for children with mental health problems in NWP.

In this study, AI was applied in building on strengths that already exist in the mental health care system to create an awareness of the positives within, and to create prospects in dealing with mentally ill children. Appreciative inquiry is one of the thinking and communication approaches about the past, current, and future state [10]. It shifts the focus from failure to team empowerment [19]. The researcher is engaged with participants through interaction and observation. Through this interaction, the researcher gained knowledge about psychiatric nurses providing mental health care services to children with mental health problems in the NWP.

The dominant approach to the AI process is the five D Model. The five key phases in this process are the Definition phase, Discovery phase, Dream phase, Design phase, and Destiny phase. Figure 1 below clarifies this process.

### 2.3. Setting

The study took place in NWP of SA across four districts in accredited psychiatric institutions where child psychiatry services are rendered. Each district runs the mental health services according to the needs and the uniqueness of the community they serve. The psychiatric services are led by the provincial Department of Health with officials trained in general health and mental health. These officials are also in district health offices with each district having at least one accredited psychiatric institution. Child psychiatric services in this province are scanty and not clearly defined. Staffing ratios could be clearly stipulated but these are not adhered to in the practical reality.

### 2.4. Data Collection

Appreciative Inquiry (AI) is a form of action research that attempts to help individuals, groups, organizations, and communities to create a new vision for themselves based on a positive understanding of their experiences [20]. In this study, AI aimed to advance and promote the positives of psychiatric nurses caring for children with mental health problems. A central premise of AI is that a deliberate positive process of knowing is constructed and created through dialogue [13]. In this study, participants were recruited from the four public health institutions where child psychiatric services are offered. Psychiatric nurses rendering child mental health services for at least six months were invited to participate in the study. Both males and females were included in the study. Those that met these inclusion criteria and volunteered to participate were included in the study. Purposive sampling method was used. Five focus group discussions were conducted across four districts in NWP where child mental health services are offered. The first author facilitated focus group discussions. Each of the focus groups (FGs) lasted approximately an hour. Conversations in the focus groups were audio-taped and transcribed verbatim. The focus groups were moderated by the study supervisors who are both experienced qualitative researchers. A total of 35 psychiatric nurses participated in focus groups. The participants consisted of five males and thirty females. The age of the participants ranged from 26–59, while five of the participants did not disclose their age. The focus group discussions were conducted from May to July 2022, when COVID-19 restrictions were lifted in South Africa. The group sessions were designed to understand the internal strengths of psychiatric nurses and the positives within the system in child mental health services.

### 2.5. Data Analysis

Data analysis involved a participative approach based on appreciative inquiry strategies. The first author had regular data analysis meetings with the two supervisors to go through data to discuss data generation, make sense of possible meanings, and identify the key themes for analysis. An independent coder who is experienced in qualitative research was also used.

### 2.6. Ethical Approval

Ethical approval to conduct the study was obtained from the Scientific Committee of the School of Nursing Science and the North-West University Health Research Ethics Committee (HREC Reference Number: NWU-00278-21-A1). This is part of a doctoral research by the first author. Permission to conduct the study was requested and obtained from the North-West Provincial Department of Health, directors of district health services of NWP, and Chief Executive Officers of accredited mental healthcare institutions. Permission was obtained from participants, and they signed informed consent forms before taking part in the study. Ethical principles of mutual respect and fairness were applied, and the authors assumed full responsibility for the research project.

## 3. Results

Data generated from the psychiatric nurses were transcribed and analyzed using the Five D model of AI, as summarized in Table 1 below.

### 3.1. Definition Phase

This AI focused on the success of psychiatric nurses’ strength and best practices in caring for children with mental health problems in NWP. There is currently no evidence in the literature that focuses on child mental health nursing in NWP. However, psychiatric nurses still need to take care of children with mental health problems due to a high demand for this service. Furthermore, there is scarce information on child mental health services in NWP. There are psychiatric nurses that are specialized in child psychiatry, even though there is a scarcity of child and adolescent psychiatrists throughout the province.

### 3.2. Discovery Phase

The discovery phase of the AI cycle in each focus group session was an initial success. This brought together psychiatric nurses describing what works in caring for children with mental health problems in their unique environment. During these group sessions, psychiatric nurses described their strengths and attributes in caring for children with mental health problems. The research question was rephrased in line with an AI approach for the Discovery phase as follows: *What works for you in caring for children with mental health problems?* A number of themes emerged, as detailed in the next subsections.

#### 3.2.1. Staff Willingness, Enthusiasm, and Commitment

Values play a central role in guiding psychiatric nursing, because these are considered worthy for the practice environment [21]. Psychiatric nurses were aware and ready to respond to the children’s needs because nursing values and an ethic of care guide their practice environment. This included commitment, i.e., enthusiasm and passion to work with children, as postulated in the following vignettes:


*FG 3 Participant 1 ‘What is working is let’s say people are committed to come to work. You come and you are committed to do your best.’*



*FG 1 Participant 1 ‘Where we are working, our staff are committed despite them not having the speciality of mental health in children. But they are committed at any point or whatever. At the end of the day, we are managing these children.’*


Adequate knowledge regarding the effect of mental illness on children may have a positive contribution to the psychiatric nurse’s willingness to work in this environment. The authors of [22] are of the view that understanding how psychiatric nurses identify and apply their professional nursing values is an important step towards improving psychiatric nursing practice and quality of care towards children with mental health problems. These values emerge out of academic preparation and socialization into the profession.

#### 3.2.2. Functional Multidisciplinary Team (MDT)

There is currently full multidisciplinary team participation in most facilities that offer child psychiatry. Even though the services are limited, and the leading doctor does not specialize in child psychiatry, psychiatric nurses and all other team members involved in the care of children have demonstrable passion in what they are doing. The following submission confirms the current clinical practice.


*FG 5 Participant 2 ‘We have a full Multidisciplinary Team.’*



*FG 3 Participant 1 ‘For me what works is multidisciplinary team collaboration. So, on Wednesdays we have what we call child clinic. So, if we have a difficult or complicated case, we get to call the family and the child as well. And then our specialist, the psychiatrist actually does the interview while the child is there. She does the assessment. So, I feel like that is working.’*


In other countries around the world, there is rapid development of disciplines and child psychiatry has reached an internationally advanced level [8]. In NWP, the one university offers degrees for basic nursing, social work, and clinical psychology in childcare. Working towards attracting more graduates into mental health is important for clinical practice. The multidisciplinary nature of child mental health could be further improved if other disciplines pay close attention to child mental health in this province [8].

#### 3.2.3. Involvement of Family Members in the Management and Treatment Plan

The primary care providers need education and training to care and understand the needs of children with mental health care challenges. These caregivers need to be educated and trained in seeking medical help early and facilitating early diagnosis and basic treatment for common mental disorders in children through participatory groups [8]. The following quotes confirm the need:


*FG 3 Participant 3 ‘I think maybe to educate the parents of the patients.’*



*FG 2 Participant 3 ‘Active involvement of family members is crucial in the care, treatment and rehabilitation of children who have been diagnosed with mental illness.’*


This involvement ensures that families are empowered to gain greater control over decisions that affect the mental health of their children. A family-centered approach can help to maintain and strengthen important family relationships, and to identify and enhance the strengths within them which all contribute to the recovery of a child with the mental illness [23]. It is important to provide family focused services by engaging both parents and children in the prevention and self-management. At the same time, this approach empowers families to raise their expectations of the mental health care systems may also be of assistance [24]. School-based programs for self-management of children with mental health problems may also be helpful. This would mean children having mental health challenges are treated in appropriate ways.

### 3.3. Dream Phase

In the dream phase, opportunities are created for the psychiatric nurses to describe an ideal service for the child psychiatric nursing in NWP. The psychiatric nurses described the ideal possibilities for the child psychiatric and mental health service. They described the best possible outcome that could work for the ideal child psychiatric service. The research questions were rephrased in line with an AI approach for the Dream phase as follows: *Where do you want to see the care of children with mental health problem at in NWP?* A number of themes emerged, as detailed in the next subsections.

#### 3.3.1. Acceptable Times: Children’s Clinic for Mental Health

Integrated care, where all care is delivered by one team in one location, or collaborative care, where primary and mental health care practitioners coordinate care through close communication, increase the likelihood that children will receive needed mental health care services [25]. This is an observation that emerged differently in the study. The following quote confirms the need for such a service:


*FG 5 Participant 1 ‘The clinic to run Monday to Friday without limitations till 4 p.m.’*


Reference [26] highlighted the known fact that critical human developmental changes take place within the first 18 years of life. Psychiatric nurses need to assess for any of the typical presenting emotional, behavioral, or developmental complaints and fill this important gap to care for this vulnerable population while striving to achieve mental health. Children’s mental health is complex in nature. Currently, in NWP, the mental health services to children are characterized by multiple silos of influence where children receive care. There have been increasing calls to integrate mental, emotional, and behavioral wellness care into the provision of general health care of children, but significant barriers remain [27].

#### 3.3.2. Improvement of Outreach Programs

Focusing on outreach programs means empowerment and unleashing the power in people through their knowledge, experience, and motivation. Through these programs, society benefits from information sharing, which may contribute towards raising the level of trust in the mental health care system in the community. Sharing information may sometimes mean disclosing information that is considered privileged or problem areas. Beyond increasing access to care, working in the community helps psychiatric nurses provide care informed by the strengths and challenges unique to the community. Outreach programs may also assist in seeking knowledge about the community by constantly scanning the environment which one is serving and transfer knowledge through encouragement of dialogue [28]. Outreach programs ensure that the community is empowered with skills and knowledge to become innovative in their thoughts and understanding of mental illness. This is confirmed by the following vignettes:


*FG 1 Participant 8 ‘Even if you can understand how to work with children, children come from different homes and from different families, where they don’t really understand when the child is having a problem. So may be strengthening mental health services at the community also needs to be attended to.’*



*FG 2 Participant 2 ‘With the community there is always a need for education, which I think NW has... should I say zero because a lot of people actually do not know. If it was not for pre-school teachers and primary school teachers, then a lot of parents would be with these children and not aware that their child is sick. They would just see the child as naughty, as different from the others, but they don’t acknowledge it as a mental problem that needs to be attended to. Hence most of the time if the child does not attend preschool, she gets to be seen in primary school. So, I think what need to be done the whole province is education. Whether that is via media, via pamphlets, whether is …you know, more focus should be taken back to mental illness not only the normal mental illness—the adult mental illness, but we are now looking at kids. Because a lot of families need it, if any one of us can get a slot on YOU FM and talk about ADHD. I am sure the following week we will be full in the clinic.’*


### 3.4. Design Phase

Once the dream phase was completed, the psychiatric nurses then identified their need for a conducive child-friendly service. This phase entails creating possible propositions. The psychiatric nurses commit the institutional managers in ensuring child psychiatric services are realized. The research question was rephrased in line with an AI approach for the Design phase as follows: *What can be done to maximize the potential in the care of children with mental health problems?* The following responses qualify their proposition:


*FG 5 Participant 4 ‘Come with money and build the hospital with staff.’*



*FG 2 Participant 1 ‘Build us a hospital. Well equipped. May be a twenty (20) bed hospital dedicated just for kids. Because that really is what they need. Like a playroom even though it is a positive thing that we do have a child clinic, the set up sometimes is disadvantageous to the kids because there is no one way mirror.’*


A number of themes emerged, as detailed in the next subsections.

#### 3.4.1. Training and Development of Multi-Disciplinary Team Members in Child Psychiatry

Staff development is an essential tool in realizing quality of care for children with mental health problems. Staff empowerment is necessary when the following indicators are clear: insufficient knowledge, reported limited work skills and abilities, unsupportive practice environment, and poor physical health [29].

Educating staff on the effect of mental illness on children may be critical for increasing willingness to participate in the care for these children and improve quality care. This may include activities that support improved quality of life for those involved in the care for children with mental health problems [30]. The following quote confirms this need:


*FG 1 Participant 3 ‘There must be a psychiatrist, a psychologist that will be able to assist with the child. A multidisciplinary team. We need a psychiatrist who is trained with children or a psychologist who is trained specifically with children because all other professions have got that.’*


Another participant submitted:


*FG1 Participant 6 ‘I would like to specialize: Training and development on mental needs especially with children. We must have more training for nurses on child mental health.’*


In the same vein, another participant in the focus group discussion had this to say:


*FG 2 Participant 3 ‘Even other things like integrating mental health into the school health program or services. Also having more training for nurses on child mental health and adult psyche. And the support and understanding of management. Because sometimes even if they can be trained you find that the managers don’t have skills and understanding. It will be just a fruitless exercise. They must have an understanding on mental health and child psychiatry.’*


Psychiatric nurses indicated willingness to learn more about children and mental illness. Characteristics associated with increased willingness were young nurses who seek to specialize in the care of children. They affirmed the importance of relevant disciplines and research in child health care, special education schools, and training centers for the successful implementation in child mental health services [8]. While the current training of a psychiatric nurse includes limited theoretical and clinical knowledge, the inclusion of exposure and comprehensive nursing programs which provide sufficient knowledge and skills to work with children who are mentally ill would be a welcome extension.

#### 3.4.2. Investment in Infrastructure

It became clear in this study that the mental health of children requires multisectoral cooperation and the attention of the whole society, as suggested and recommended by [8]. Despite some health gains over the past 28 years in SA, children are not reaching their health potential in many rural areas and in low-income and middle-income families. Structural reforms and redesigning service delivery to maximize outcomes are more likely to improve service quality [24]. The participants confirmed the need to invest in infrastructure as exemplified through the following quotes:


*FG 2 Participant 2 ‘And a child friendly clinic as well. Because some children still make it as an outpatient client. But they would need…Even our psychologist I think they are doing their best right now but, they could actually do better if they could have play therapy equipment, a playroom, you know like a garden outside to play and see how they respond.’*



*FG 1 Participant 4 ‘If they can build psychiatric hospitals in rural areas.’*


According to van Rensburg [31], the South African Society of Psychiatrists (SASOP) has called for a complete overhaul of the mental healthcare system following the Health Ombudsman’s report on the deaths of psychiatric patients in the Life Esidimeni incident in 2012. The deaths provoked a national outcry. This rectification order from the Ombudsman includes the identification and costing of the required facility and staffing interventions that must be put into place to ensure capacity and integration at the different service levels. There have been calls to provinces including NW to ensure the actual implementation of mental health policy and to urgently translate it into practical action plans, with enabling budgets to fund the required facilities and staffing. In this way, facilities at the different levels must be brought up to an acceptable standard, including identified projects at acute specialist units and 72-h assessment facilities at district hospitals. This is still to be realized in NWP.

### 3.5. Destiny Phase

This phase entails strengthening the system’s affirmative capability, and thus, building hope for ongoing positive change and high performance for sustainability. Delivery of this dream must take place. One theme emerged, as described in the following subsection.

#### Availability and Upgrading of Infrastructure

Infrastructure plays an important role in the delivery of appropriate health service to everyone in need of it. Reference [32] highlighted that the health system infrastructure includes physical facilities, information systems, and material and human resources and may involve the construction of new infrastructure as a strategic goal achievement. According to [33], upgrades to the infrastructure can lead to improved health outcomes and help achieve the MDGs. Availability of infrastructure and upgrading of it to support safe children’s mental health services in health care facilities is an important step in the journey towards achieving National Health Insurance (NHI) in NWP and an important component of a well-functioning healthcare system [32]. The discussions in this study highlighted concerns regarding infrastructure to meet the mental health needs of children. The research question was rephrased in line with an AI approach for the destiny phase as follows: *How can we help in caring for children with mental health problems?* The following response affirmed the desired destiny:


*FG 1 Participant 4 ‘If they can make a ward and a child mental health specialized one, the one which specializes with kids. And for MDT to take place as well.’*


Another participant added the following:


*FG 2 Participant 1 ‘If it was a fantasy world, where we could have a 20-bedded, dedicated child unit, then obviously it would be best to separate them so that you go in depth.’*


In the current and uncertain healthcare climate, access and uniformity remain significant barriers for a large portion of communities in NW province. Many issues impact the mental health of children today and these, consequently, create challenges for both families and communities [26]. There is a need for basic and appropriate infrastructure to service the needs of children with mental health challenges. Provision of such services and resources would enable nurses to offer quality services to those who need them. Generally, children want to connect with their peers. Therefore, pathways to care should be embedded within the wider community to ensure easy access for individuals and their families [34]. Such services would alleviate suffering and aid recovery to minimize complications that may arise in later stages of life. This is confirmed by the following:


*FG 1 Participant 6 ‘Even if [it means] we can have a child psychiatric clinic in each village where it becomes easier to communicate.’*


The last of the participants added the following:


*FG 2 Participant 3 ‘Even if they build this small hospitals or psychiatric clinic. It will help.’*


## 4. Discussion

Psychiatric nurses are clinical mental health care providers available for patient care 24 hours a day. In primary health care settings, they provide mental health care, treatment, and rehabilitation services, including diagnosis and management of both acute and chronic mental illness, as stipulated in Mental Health Care [35]. These services are also provided for and prescribed in health establishments, such as community health and rehabilitation centers, clinics, and general and psychiatric hospitals. For these services to be realized, a coordinated and objective planning must take place. Reference [36] highlighted that planning as a process gives effect to the wishes of the community. It is also a process in which a community is organized to recognize its needs, and to institute means for their satisfaction. A key factor in the success of child psychiatric services in NWP is the involvement of the provincial mental health leaders and district teams throughout the planning and implementation process. They must participate fully. Structural reforms to address the deficits in NWP ’s health system is more likely to improve service quality for children with mental health challenges. Mental health systems are important for children’s health and emotional wellbeing. The process described in the next section may assist in realizing the discovered dreams, as postulated by [37].

### 4.1. Basic Belief and Values

The nursing profession has values and belief systems which are significantly correlated, and psychiatric nurses conform to them. Self-awareness and compassionate care are regarded as important values for developing an effective nurse–patient relationship and for improving nursing abilities. They are both regarded as important components of nursing by many scholars in the nurse–patient relationship focus area. This means that being self-aware is a combination of meeting the complex needs of the patient and intrapersonal and interpersonal communication [38]. Psychiatric nurses are required to maintain safe and optimal participation in daily nursing activities in a variety of clinical contexts. Becoming self-aware enables nurses to be fully present for children with mental health challenges [39]. The presence of this value system has positive impact on the psychiatric nurses’ personal and professional development. The expansion of this area will increase the level of confidence among individual psychiatric nurses or between team members, including commitment. This will assist nurses in building resilience and success in this field.

Health care providers are generally expected to respond to calls for enhancing compassionate care. This requires psychiatric nurses to be sensitive to children with mental health challenges and have sufficient knowledge to make critical clinical judgments in order to take care of these children [40]. Current literature lacks insight in demonstrating the actual impact that self-compassion has on the recipients of compassionate care, despite this being highlighted as a central outcome of self-compassion interventions [40].

### 4.2. Changing Operational Practices

In order to efficiently acquire the knowledge necessary for decision making at the clinical setting, including the field of psychiatric nursing, the NWP department of health needs a system of knowledge management, information systems, and databases related to psychiatric nursing. Based on the challenges phased by psychiatric nurses in this province, the establishment of organizational cultural development and support strategies should be considered [41]. Interestingly, people with high motivation and commitment remain optimistic even when the results are against them. Training and reflection on how child psychiatry practice should be introduced and integrated into the clinical practice of these nurses to suit their workplace context must be considered. Encouraging teamwork and continuing professional developmental opportunities to improve psychiatric nurses’ knowledge, skills, and abilities can be used as a strategy. A mentoring program may also be considered.

Efforts must be made to reorganize suitable psychiatric multidisciplinary teams in order to collectively contribute to the child psychiatric practice environment in North West Province in South Africa. This include socially skilled individuals who are skillful in managing teams. However, the reality is that the COVID-19 pandemic came with challenges to the nursing practice environment. More emphasis needs to be placed on matters associated with the design of collaborative and collective work patterns, team performance and motivation, psychiatric nursing staff stress levels, and their continuous development to be up to date with the issues affecting the clinical practice [41].

### 4.3. Strategy and Tasks

The literature suggests that psychiatric nurses may draw on normalization process theory (NPT) to examine how changing operational practices can be rooted within their clinical setting and practice environment. NPT identifies clear factors that promote and prevent the routine incorporation of complex interventions into everyday practice [42]. These include the following construct: coherence, cognitive participation, collective action, and reflexive monitoring, which are discussed below

Coherence: this refers to the process of gathering information which the psychiatric nursing practice undergo in order to understand their clinical practice. For example, this allows them to define and examine the implications of the decisions on defining and re-organizing the psychiatric nursing practice.

Cognitive Participation: this examines how the other participants, including multidisciplinary team members, engage in the newly adopted child psychiatric practice. This allows for all the stakeholders to identify the psycho-social, clinical roles and responsibilities which are developed to sustain change in the child psychiatric practices of NWP.

Collective Action: this focuses on the nursing care which psychiatric nursing team ought to do to change the child psychiatric clinical practice by endorsing the new practices. This allows psychiatric nurses to examine the specific nursing care and practices, organizing factors, and new means that can be used to sustain child psychiatric clinical practices facilitated by multidisciplinary team working towards the same vision.

Reflexive Monitoring: this describes the value realization inherent in the informal and formal appraisal of the psychiatric nursing practices and the reported improvements in the nursing care of children with mental health challenges. This can also provide new insights on the effectiveness of the psychiatric nursing care forging new mental health care directorate structures in the province.

### 4.4. Leadership and Decision-Making Processes

The current literature constructs view leadership as a process of influence which coordinates change processes and knowledge management. Leadership encourages decision making based on available information and fosters communication among staff members for informed practices and processes. At the same time, knowledge management techniques help to provide consistency of services, regardless of the physical location of the employee. Sharing information throughout the clinical practice is a key pattern for knowledge management in the NWP’s health department [43].

An inclusive approach to decision making is seen as important for both patient outcomes and the engagement of psychiatric team members. Collaborative decision making is central to building confidence in psychiatric nurses. It is also important that the responsibility as a decision maker is taken seriously and that the ultimate decisions about child psychiatric services in NWP is such that it could have a positive impact on the public safety and the safety and wellbeing of children as service users [44].

Research has indicated that many young people worldwide are not well informed about mental health and there is a clear need to raise awareness, educate, and provide interventions that facilitate the maintenance of mental well-being in children and adolescents. Mental health promotions are potentially central to the solution, and therefore, it is unsurprising that many interventions that take this approach have been developed and are successful [45]. Reference [24] postulated that children and adolescents are not reaching their health potential in many low-income and middle-income countries. The situation becomes more complex in rural and poor communities. The same paper indicated that the quality of services for children is substandard across both health and social systems. By telling successful stories during the focus group discussions, psychiatric nurses recognized their potential and inner strength.

### 4.5. Implications for Practice

Psychiatric nurses, nursing educators and the hospital managers must advocate for an increase in the presence of psychiatric nurses in clinical settings besides mental health institutions where their skills are needed. These include community health care centers and clinics and emergency and casualty departments, which often serve children and adolescents with medical and psychiatric comorbidities. Children’s needs must be met through positive nursing values. This will enhance quality mental health care that is safe, effective, timely, efficient, equitable, and child centered, as indicated by WHO [46]. Children’s mental health is complex in nature. Currently, in NWP, the mental health services provided to children is characterized as multiple silos of influence. There have been increasing calls to integrate mental, emotional, and behavioral wellness care into the provision of general health care of children since the Life Esidemeni deaths in 2017, but significant barriers remain [31]. Raising awareness about the plight of children’s mental health is equally critical. The discussions detailed in this paper generates multiple principles that are fundamental and essential for the success of psychiatric nursing practice. Such evidence is needed to ensure that psychiatric nursing practice remains relevant to children with mental health challenges and mental health care practitioners by providing evidence on what works best. Understanding child psychiatric nursing practice through problem-based approaches may not be helpful. The results of this study could be used to launch targeted initiatives to improve the quality of care in child psychiatric settings or children with mental health problems. This AI sets a new standard in the practice environment of psychiatric nursing in NW province.

### 4.6. Limitations

This study took place in different child psychiatric settings, so it is important to acknowledge the limitation of its context for transferability. It is also a small-scale study, with an absence of prior research conducted on child psychiatric nursing in NWP which it can be compared to. The social and physical environments of the institutions offer a range of constraints to psychiatric nursing practices and experiences. Readers need to consider how the knowledge generated in this study may be applicable to their own settings and decide how they may adjust and embrace this knowledge.

## 5. Conclusions

Appreciative Inquiry (AI) verified the commitment of psychiatric nurses in caring for children with mental health problems and the potential for dedicated child psychiatric institutions in realizing the needs of such children in NWP. Even though the structural reforms are more likely to improve the quality of care of children’s mental health, a clear vision is needed to guide efforts within the mental health care system in NWP, which may contribute to the adequacy of care received by children. The improvement and establishment of dedicated children’s mental health services in both primary health care and school health services are critical components in rebuilding the mental health system for children in NWP. Research advances on the progress of nursing care, treatment, and rehabilitation services for children with mental health challenges in NWP is needed.

## Figures and Tables

**Figure 1 ijerph-20-01725-f001:**
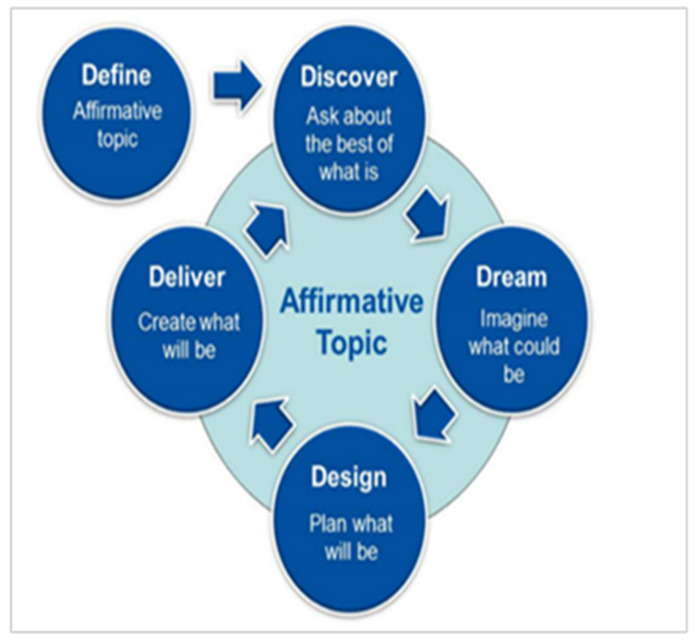
The Appreciative Inquiry (AI) five D model.

**Table 1 ijerph-20-01725-t001:** Summary of the five phases of Appreciative Inquiry.

	Research Question	Themes
Definition phase	None	Not applicable
Discovery phase	What works for you in caring for children with mental health problems?	Staff willingness, enthusiasm and commitment Functional multi-disciplinary team (MDT) Involvement of family members in the management and treatment plan
Dream phase	Where do you want to see the care of children with mental health problem in NW province?	Acceptable times for children’s clinic for mental health Improved outreach programs
Design phase	What can be done to maximize the potential in the care of children with mental health problems?	Training and development of MDT members in child psychiatry Attention to infrastructural issues and staff shortage
Destiny phase	How can we help in caring for children with mental health problems?	Upgrading of infrastructure

## Data Availability

Not applicable.

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
