# Peer review of "Towards an Understanding of Successes of the Psychiatric Nurses in Caring for Children with Mental Health Problems: An Appreciative Inquiry"

_ijerph, 2023, doi:10.3390/ijerph20031725_

Round 1

Reviewer 1 Report

Dear authors, congratulations for this important research and for the manuscript presented. Some suggestions to improve it: 

a) Don't need to put the pages after the reference presented (e.g. line 35" [3] (p.44)". You repeat tthe reference [3] so it is understandable that the information is in different parts of the reference. Check all these mentioned pages in your manuscript. 

b) Line 46 Instead of " (2016) defines success", use the name of the author(s). "Onyibo defines success". And check the reference because is not complete. 

c) In line 54 are you sure that "The mental health care system in NorthWest province is nurse-based because the geospatial characteristics are largely rural." That's the main reason? 

d) In line 107 check the sentence: "Notwithstanding, [13] (p.2) point to [14] (2017) who indicate that AI supports learning and reflection in a positive way." it can be improved to a better understanding. 

e) I believe you don´t need to explain so much information about AI. Just the enough information to justify the choice of the methodological approach. 

f) You don't mention anywhere clearly how you reach the 35 psychiatric nurses that participated in your study. Please include these informations.

g) In line 191, there is too much spce between the words. check formating: 

h) Identify the codes of the sentence makers (e.g. Line 196). 

The rest is ok!  Just a litle more effort to improve your manuscript

Author Response

Dear Reviewer

Thank you for  reviewing my manuscript and the comments.

Please see attached rebuttal documents

Reviewer 2 Report

Thank you very much for the opportunity to review the article. 

Congratulations to the authors because they did very interesting research and wrote an interesting article. 

In the spirit of AI, the authors interestingly Designed the article, then Delivered it. As a reviewer, I Discovered it, and now both the authors and I Dream of making it the best possible. 

To this end, I send below some comments, insights, and suggestions on the article. 

Changes in the Introduction, the methodological chapter and Discussion

1.     One important note: in the Introduction, the authors use the abbreviation AI, but before that (except for the Abstract), there was no development of this abbreviation. This needs to be corrected - as it is only in line 101

2.     In connection with the fact that in line 87, the authors mention AI, in my opinion, it would be worthwhile in the Introduction to add a subsection, e.g., Appreciative Inquiry Assumptions, and insert what is in the chapter Materials and Methods lines: 101-120. 

3.    In the Materials and Methods chapter, it is worth starting with the Study Design subsection and briefly describing in it why AI is included in action research AND what AR is characterized by - you can use this, for example, the following articles and connect it to what's already in lines: 122-135

Appreciative Inquiry: An Action Research Method For Organizational Transformation and its Implications to the Practice of Group Process Facilitation by Troxel

Appreciative Inquiry as a mode of action research for community psychology by Boyd

4.    The logical order of the subsections in Materials and Methods could be as follows:

Study Design

Settings

Data Collection

Ethical approval

5.    The Data Collection lacks information on how nurses were invited to participate in the study. Was it a purposive selection? If so, what did it consist of? Was it a random selection? If so, for what reason did the authors decide on it?

6.    There is also a need for more information about the focus groups' conduct. How long did they last? Who moderated them? Did the same people moderate all the focus groups? Were they the authors of the study? If not, how did the study authors prepare them? What was the experience of these people in working with the AI method? How, and based on what was the focus scenario constructed? Did all the focus groups proceed according to the scenario? From further work, there were prepared questions for each phase of the model. If so, this should be described in the Materials and Methods section. 

7.     How did the authors analyze the collected data? Did they use any software?

8.    The discussion needs serious supplementation. Only two sentences (lines 432-437) refer to the research described above. Applying the AI method requires bringing to completing the task at hand. The method involves not only describing the way things are but pointing out possible fields of development. The discussion also requires a deeper reference to the success that the authors wrote about in the Introduction and to see if the determinants of success mentioned in the Introduction are similarly perceived by the respondents who participated in the focus groups. If so, to what extent? If not, what new contributions did the surveyed nurses make? 

9.    In the authors' opinion, did the study have any limitations? The use of AI brings limitations in looking at the phenomenon under study. 

Minor comments

·      I need help understanding the given in parentheses (p.24) next to each reference. What purpose does this serve?

·      Line 156: a sentence should not begin with a reference

·      Lines: 214, 216 Multidisciplinary or Multidisciplinary

·      In lines: 198, 215, 234, in each case, the hyphen "and" are written differently; on this occasion, I have a question, does "and" mean that the same person is saying it?

·      Lines 199, 216, 214 sometimes quote from what the research said to start with quotation marks, sometimes not.

·      Line 467: it makes sense to use one notation Appreciate Inquiry and not Appreciate Inquiry

Author Response

Dear Reviewer

Thank you for reviewing my manuscript and comments that edify. Please see attached rebuttal document

Round 2

Reviewer 1 Report

Dear authors, thank you for your improvements. The manuscript is now much better. 

But regarding the suggestion: " Identify the codes of the sentence makers (e.g. Line 196)." you told in your answer that: "codes have been identified infront of each quote". But in the version submited in the system the codes still don't appear. If you can put it it can be really better! 

Thank you! 

Author Response

Codes have been identified in front of each quote

Reviewer 2 Report

Congratulations to authors!

Thank you.

Kind regards.

Author Response

The authors' affiliations have been indicated.
